# Bright-Dark and Multi Solitons Solutions of (3 + 1)-Dimensional Cubic-Quintic Complex Ginzburg–Landau Dynamical Equation with Applications and Stability

**DOI:** 10.3390/e22020202

**Published:** 2020-02-10

**Authors:** Chen Yue, Dianchen Lu, Muhammad Arshad, Naila Nasreen, Xiaoyong Qian

**Affiliations:** Faculty of Science, Jiangsu University, Zhenjiang 212013, China; ychen108@uncc.edu (C.Y.); naila_nasreen@hotmail.co.uk (N.N.); 5853868@163.com (X.Q.)

**Keywords:** modified extended simple equation and exp(−*ϕ*(*ξ*))-expansion methods, proposed F-expansion method, cubic-quintic complex Ginzburg–Landau equation, multi solitons, periodic solutions, solitary wave solutions

## Abstract

In this paper, bright-dark, multi solitons, and other solutions of a (3 + 1)-dimensional cubic-quintic complex Ginzburg–Landau (CQCGL) dynamical equation are constructed via employing three proposed mathematical techniques. The propagation of ultrashort optical solitons in optical fiber is modeled by this equation. The complex Ginzburg–Landau equation with broken phase symmetry has strict positive space–time entropy for an open set of parameter values. The exact wave results in the forms of dark-bright solitons, breather-type solitons, multi solitons interaction, kink and anti-kink waves, solitary waves, periodic and trigonometric function solutions are achieved. These exact solutions have key applications in engineering and applied physics. The wave solutions that are constructed from existing techniques and novel structures of solitons can be obtained by giving the special values to parameters involved in these methods. The stability of this model is examined by employing the modulation instability analysis which confirms that the model is stable. The movements of some results are depicted graphically, which are constructive to researchers for understanding the complex phenomena of this model.

## 1. Introduction

The nonlinear Schrödinger’s equations (NLSEs) are well-known models in nonlinear partial differential equations (PDEs) to govern the optical soliton propagation [1,2,3,4,5,6,7,8,9,10,11,12,13]. Since the 19th century, many researchers have focused on the PDEs with complex variables [14]. Succi was the pioneer who proposed the complex distribution of the Gross–Pitaevski model through utilizing a lattice Boltzmann model for Bose–Einstein condensation in 1998. After that, expanding this model to a higher-dimensional equation was described [15,16]. The authors in [17] proposed one more complex division for NLSE wherein the function of the wave is separated into phase angle and amplitude.

From NLSEs, the cubic-quintic complex Ginzburg–Landau model is a continuing estimate to the dynamics of the area in a passively mode-locked laser. It is admitted that this model is constructive in demonstrating imperative phenomena—for example, erbium-doped fiber amplifiers and propagation of ultra-short pulses in optical transmission lines having spectral filtering. In physics, the generalized quintic complex Ginzburg–Landau equation (GQCGLE) was utilized fruitfully in the formation of other non-equilibrium procedures. In the model, the quintic term explains the physical significance that is absent in existing models in a previous study [18]. It occurs in various branches of science and plays an important role in practical applications including fluid dynamics, nonlinear optics, mathematical biology, hydro dynamical stability problems, condensed matter physics, Bose–Einstein condensates, chemical reactions, super conductivity, and quantum field theories [19,20,21,22,23]. It demonstrates prosperous dynamics and has turned into an example for alterations to the chaos of spatio-temporal. In Refs. [24,25], some other aspects regarding the various forms of Ginzburg–Landau models are described. This model as well occurs in the investigation of chemical systems studied via reaction–diffusion equations. The GQCGLE Model also plays the part of a simplified form in fluid dynamic equations. Recently, numerous researchers have concentrated on this model that involve localized complex forms in optical media—for example, necklace–ring solitons, vortex solitons, and so on [26,27,28].

Complex Ginzburg–Landau models of higher dimensions from complex lattice Boltzmann models were proposed by the authors [29,30,31]. These problems were built on the complex lattice Boltzmann equations. In physics and applied mathematics, the different forms of complex Ginzburg–Landau equations have attained great attention from researchers, as universal models where the most interesting solutions are dissipative solitons. In optical lattices of nonlinear self-localized structures, one- and two-dimensional arrays frequently assigned as discrete solitons have been forecasted and examined [32]. In [20], mathematical and physical features of the equations based on GQCGLE were reviewed. As numerical tools, the lattice Boltzmann method (LBM) has been broadly used in several study areas for examining fluid dynamics in the previous decade. The key theme behind using a lattice Boltzmann technique is to implement a straightforward mesoscopic equation of a fluid flow, generally concerning some distinct particle velocities that are not enough for describing properly the macroscopic flow pattern as the macroscopic PDEs are improved from the mesoscopic equation preserving preferred physical quantities and the principal to accurate fluxes of the preserved quantities [33,34,35]. Different self-interaction potentials were engineered in order to obtain nontrivial analytic results, with the purpose of testing the robustness of regarding the soliton as the ground state of the hairy sector, and its key role in the microscopic counting of hairy black hole entropy [36].

In recent years, NLSEs have attracted much attention from the constructing solitons and numerical solutions due to them being widely used to explain nonlinear complex phenomena. Several conventional techniques are extracted to obtain exact solutions such as the Tanh and Sech methods, the inverse scattering method, the extended tanh method, the Hirota bilinear and Darbox transform methods, the Bäcklund transform method, the generalized F-expansion technique, the Jacobi elliptic function expansion technique, the reduced differential transform method, the modified direct algebraic technique, variational iteration methods, and several others [28,29,30,31,32,33,34,35,37,38,39,40,41]. The authors in [40] have been working on CGLEs using different techniques like the modified simple equation method. Inspired from the above works, we intend to construct a new soliton solution of the CQCGLE model using a modified extended simple equation method. The soliton profile is sensitive to entropy, i.e., due to the changes in the entropy amplitude and the width of solitons. It is also observed that the increasing ion temperature and increasing magnetic field affect the shape of the soliton.

In the current paper, the aim is to construct the novel analytical solutions in different forms such as dark-bright solitons, multi solitons, kink and anti-kink waves, solitary wave, and trigonometric function solutions of (3 + 1)-dimensional CQCGLE via the modified extended simple equation method (MESEM), exp(−ϕ(ξ))-expansion method and proposed F-Expansion method. The achieved solutions are exact and more general. The modulational instability (MI) analysis is discussed by employing standard stability analysis that confirms that the model is stable and the achieved solutions are also stable.

This paper is organized as follows. An introduction is given in Section 1. The mathematical model is described in Section 2. The important steps of the projected techniques are explained in Section 3. Section 4 applies the proposed methods on (3 + 1)-dimensional CQCGLE. In Section 5, the stability of this model is discussed by using modulation instability analysis. The discussion of results and their Physical Interpretations are explained in Section 6. The conclusions are revealed in Section 7.

## 2. Mathematical Model

The (3 + 1)-dimensional CQCGLE [42] has the form of
(1)$$i\frac{\partial q}{\partial z}+\left(\frac{1}{2}-i\eta \right)\left(\frac{\partial^{2}q}{\partial x^{2}}+\frac{\partial^{2}q}{\partial y^{2}}+\frac{\partial^{2}q}{\partial t^{2}}\right)+i\gamma q+\left(1-i\lambda \right){\left|q\right|}^{2}q-\left(\nu -i\mu \right){\left|q\right|}^{4}q=0.$$
This model involves the derivative $\frac{\partial }{\partial z}$ which is spatial, and the system evolution depends on the *z* coordinate. The λ is the coefficient that accounts for the linear gain (λ>0) or loss (λ<0), η shows effective diffusion, λ is used here for the cubic gain coefficient, and μ accounts for the coefficient of quintic-loss. All parameters in this model are positive for knowing the physical meaning of these coefficients. In Equation (Equation 1), the term $\gamma q+\left(1-i\lambda \right){\left|q\right|}^{2}Q-\left(\nu -i\mu \right){\left|q\right|}^{4}q$ can be observed as source term.

## 3. Elucidation of Proposed Methods

In this portion, we will explain the algorithm of MESEM and the exp(−ϕ(ξ))-expansion scheme for seeking the solitons solutions of nonlinear PDEs in this section. We suppose a common nonlinear PDE, say in four independent variables x,y,z and *t*, is
(2)$$F\left(q,q_{t},q_{x},q_{y},q_{z},q_{tt},q_{xx},q_{yy},q_{zz},\cdots \right)=0,$$
where *F* is a polynomial function with respect to some specific variables that involve nonlinear and linear terms of advanced order derivatives, and the q(x,y,z,t) is an unknown function. By utilizing transformation, the q(x,y,z,t) can be converted to a polynomial function through which the complex variable can be combined into real variables.

Consider the transformation for reducing independent variables into a unique variable as
(3)$$q\left(x,y,z,t\right)=\psi \left(\xi \right),\;\;\;\xi =k_{1}x+k_{2}y+k_{3}z+\omega t,$$
where $k_{1},k_{2},k_{3}$ are different wavelengths and ω shows frequency. Through employing transformation (Equation 3), Equation (Equation 2) is transformed into ODE as
(4)$$G\left(\psi ,\psi^{\prime },\psi^{\prime \prime },\psi^{\prime \prime \prime },\cdots \right)=0,$$
where *G* is a polynomial and “$\left({}^{\prime }\right)$” indicates derivative of ψ with respect to ξ.

### 3.1. Modified Extended Simple Equation Technique

The main steps of proposed technique are:

**Step 1:** We suppose the solution of Equation (Equation 4) has the form of:(5)$$\;\psi \left(\xi \right)=\sum_{i=0}^{N}A_{i}{\left(\frac{\phi^{\prime }}{\phi }\right)}^{i}+\sum_{j=0}^{N-1}B_{j}{\left(\frac{\phi^{\prime }}{\phi }\right)}^{j}\left(\frac{1}{\phi }\right),$$
where $A_{i}$ and $B_{j}$ are real constants to be determined later. In addition, ϕ(ξ) satisfies the below second order equation
(6)$$\phi^{\prime \prime }\left(\xi \right)+\Omega \phi \left(\xi \right)=\rho ,$$
where Ω and ρ are arbitrary constants. Equation (Equation 6) gives three kinds of general solutions having two arbitrary parameters as follows:(7)$$\phi \left(\xi \right)=\left\{\begin{array}{c} A_{1}cosh\left(\sqrt{-\Omega }\xi \right)+A_{2}sinh\left(\sqrt{-\Omega }\xi \right)+\frac{\rho }{\Omega },\;\;\Omega <0,\\ A_{1}cosh\left(\sqrt{\Omega }\xi \right)+A_{2}sinh\left(\sqrt{\Omega }\xi \right)+\frac{\rho }{\Omega },\;\;\Omega >0,\\ \frac{\mu }{2}\xi^{2}+A_{1}\xi +A_{2},\;\;\;\Omega =0. \end{array}\right.$$
(8)$${\left(\frac{\phi^{\prime }}{\phi }\right)}^{2}=\left\{\begin{array}{c} \left(\Omega A_{1}^{2}-\Omega A_{2}^{2}-\frac{\rho^{2}}{\Omega }\right){\left(\frac{1}{\phi }\right)}^{2}-\Omega +\frac{2\rho }{\Omega },\;\;\;\;\Omega <0,\\ \left(\Omega A_{1}^{2}+\Omega A_{2}^{2}-\frac{\rho^{2}}{\Omega }\right){\left(\frac{1}{\phi }\right)}^{2}-\Omega +\frac{2\rho }{\Omega },\;\;\;\;\Omega >,0\\ \left(A_{1}^{2}-2\Omega A_{2}\right){\left(\frac{1}{\phi }\right)}^{2}+\frac{2\rho }{\Omega },\;\;\;\;\Omega =0, \end{array}\right.$$
where the constants $A_{1}$ and $A_{2}$ are arbitrary.

**Step 2:***N* is determined via utilizing the balancing principle among the terms of highest order derivative and the nonlinear in Equation (Equation 4), and is positive.

**Step 3:** Putting Equation (Equation 5) along with Equation (Equation 6) into Equation (Equation 4) and equating the coefficients of different powers of $\frac{1}{\phi^{\iota }\left(\xi \right)}$ and $\left(\frac{\phi^{\prime }\left(\xi \right)}{\phi (\xi )}\right)\;\left(\frac{1}{\phi^{\iota }\left(\xi \right)}\right)$ to zero capitulates a set of equations. By using Mathematica software, these sets of equations are resolved, and then the value of parameters are obtained.

**Step 4:** Substituting the values of parameters obtained in Step 3 and general solutions (Equation 7) of Equation (Equation 6) into Equation (Equation 5), solutions of Equation (Equation 3) can then be obtained in a concise way.

### 3.2. Exp(−ϕ(ξ))-Expansion Method

The main steps of this technique are:

**Step 1:** We suppose the solution of Equation (Equation 4) has the form:(9)$$\;\psi \left(\xi \right)=\sum_{i=-N}^{N}B_{-i}{\left(\exp\left(-\phi \left(x\right)\right)\right)}^{i},$$
where $B_{i}$ are real constants to be determined later. In addition, ϕ(ξ) satisfies the below second order equation
(10)$$\phi^{\prime }\left(\xi \right)=\exp\left(-\phi \left(x\right)\right)+a\exp\left(\phi \left(x\right)\right)+b,$$
where *a* and *b* are arbitrary constants. Equation (Equation 6) gives five kinds of general solutions as

**Type I** When $a\neq 0,b^{2}-4a>0$,
(11)$$\phi \left(\xi \right)=\ln\left(-\frac{\sqrt{b^{2}-4a}\tanh\left(\frac{\sqrt{b^{2}-4a}}{2}\left(\xi +c\right)\right)+b}{2a}\right).$$

**Type II** When $a\neq 0,b^{2}-4a<0$,
(12)$$\phi \left(\xi \right)=\ln\left(\frac{\sqrt{4a-b^{2}}\tan\left(\frac{\sqrt{4a-b^{2}}}{2}\left(\xi +c\right)\right)-b}{2a}\right).$$

**Type III** When $a=0,b\neq 0\;\&\;b^{2}-4a>0$,
(13)$$\phi \left(\xi \right)=-\ln\left(\frac{b}{\exp(b(\xi +c))-1}\right).$$

**Type IV** When $a\neq ,b\neq 0\;\&\;b^{2}-4a=0$,
(14)$$\phi \left(\xi \right)=\ln\left(-\frac{2b(\xi +c)+2}{b^{2}\left(\xi +c\right)}\right).$$

**Type V** When $a=0,=0\;\&\;b^{2}-4a=0$,
(15)ϕ(ξ)=ln(ξ+c).

**Step 2:***N* is determined via using the balancing principle among the terms of highest order derivative and the nonlinear in Equation (Equation 4), and is positive.

**Step 3:** Substituting Equation (Equation 9) along with Equation (Equation 10) into Equation (Equation 4) and equating the coefficients of different powers of ${\left(\exp(-\phi (x))\right)}^{i}$ to zero capitulates a set of equations. By using Mathematica software, these sets of equations are resolved, and then the value of parameters are obtained.

**Step 4:** Substituting the values of parameters obtained in Step 3 and general solutions (Equation 11) to (Equation 15) of Equation (Equation 10) into Equation (Equation 9), solutions of Equation (Equation 3) can be then be obtained in a concise way.

### 3.3. Proposed F-Expansion Method

The main steps of proposed technique are:

**Step 1:** We suppose the solution of Equation (Equation 4) has the form:(16)$$\;\psi \left(\xi \right)=\sum_{i=0}^{N}A_{i}{\left(a+F(\xi )\right)}^{i}+\sum_{j=-1}^{-N}B_{-j}{\left(a+F(\xi )\right)}^{j},$$
where $A_{i},B_{-j}$ and *a* are real constants. In addition, F(ξ) satisfies the ODE
(17)$$F^{\prime }\left(\xi \right)=d_{0}+d_{1}F\left(\xi \right)+d_{2}F^{2}\left(\xi \right)+d_{3}F^{3}\left(\xi \right),$$
where $d_{i}\;i=0,1,2,3$ are arbitrary constants.

**Step 2:***N* is a positive integer which is determined via utilizing the balancing principle among the terms of highest order derivative and the nonlinear in Equation (Equation 4).

**Step 3:** Putting Equations (Equation 16) along with (Equation 17) into (Equation 4) and equating the coefficients of different powers of $\frac{F^{j}\left(\xi \right)}{{\left(a+F(\xi )\right)}^{k}}$ to zero capitulates a set of equations. By using Mathematica software, these sets of equations are resolved, and then the value of parameters are obtained.

**Step 4:** Substituting the values of parameters obtained in the previous step and solutions of Equation (Equation 17) into Equation (Equation 16), solutions of Equation (Equation 3) can then be obtained.

## 4. Solitons Solutions of (3+1)-Dimensional Cubic-Quintic Complex Ginzburg–Landau Equation

In this part, we will apply modified extended SEM and exp(−ϕ(ξ))-expansion scheme to get a soliton solution. As the Equation (Equation 1) is complex, assume the solution in a traveling wave form as follows:(18)$$q\left(x,y,z,t\right)=\psi \left(\xi \right)e^{iP},\;\;\;\;P=\alpha_{1}x+\alpha_{2}y+\alpha_{3}z+\tau t+\epsilon .$$
Substitute Equation (Equation 18) into Equation (Equation 1) and make separate imaginary and real parts as
(19)$$\begin{array}{cc} \; & \left(\tau^{2}+\alpha_{1}^{2}+\alpha_{2}^{2}+2\alpha_{3}\right)\psi -2\psi^{3}+2\nu \psi^{5}-4\eta [\left(\tau \omega +k_{1}\alpha_{1}+k_{2}\alpha_{2}\right)\psi^{\prime }-\left(\omega^{2}+k_{1}^{2}+k_{2}^{2}\right)\psi^{\prime \prime }=0. \end{array}$$
(20)$$\begin{array}{c} \left(-2\gamma -2\eta \tau^{2}-2\eta \alpha_{1}^{2}\right)\psi -2\eta \alpha_{2}^{2}+2\lambda \psi^{3}-2\mu \psi^{5}-\left(2\tau \omega +2k_{3}+2k_{1}\alpha_{1}+2k_{2}\alpha_{2}\right)\psi^{\prime }+2\eta \left(\omega^{2}+k_{1}^{2}+k_{2}^{2}\right)\psi^{\prime \prime }=0 \end{array}$$
Using Equation (Equation 20) in Equation (Equation 19), we have
(21)$$\begin{array}{c} \left(\left(\tau^{2}+\alpha_{1}^{2}+\alpha_{2}^{2}+2\alpha_{3}\right)-4\nu \left(\gamma +\eta \tau^{2}+\eta \alpha_{2}^{2}\right)\right)\psi -4\eta \left(\left(\tau \omega +k_{1}\alpha_{1}+k_{2}\alpha_{2}\right)+\left(\tau \omega +2k_{3}+2k_{1}\alpha_{1}+2k_{2}\alpha_{2}\right)\right)\psi^{\prime }.\\ -\left(\left(\omega^{2}+k_{1}^{2}+k_{2}^{2}\right)-4\nu \eta \left(\omega^{2}+k_{1}^{2}+k_{2}^{2}\right)\right)\psi^{\prime \prime }-\left(2+4\nu \lambda \right)\psi^{3}=0. \end{array}$$

### 4.1. Solitons Solutions by Extended SEM

In this sub-part, extended SEM is employed to construct the solution of Equation (Equation 1). Apply a homogeneous balance principle on Equation (Equation 21) and assume the solution as
(22)$$\psi \left(\xi \right)=A_{0}+A_{1}\frac{\phi^{\prime }\left(\xi \right)}{\phi (\xi )}+B_{0}\frac{1}{\phi (\xi )}.$$
Substituting Equation (Equation 22) along Equation (Equation 6) in Equation (Equation 21) and setting the coefficients to zero of ${\left(\frac{\phi^{\prime }\left(\xi \right)}{\phi (\xi )}\right)}^{j}\frac{1}{\phi^{k}\left(\xi \right)}$, we obtain a set of equations in parameters $B_{0},A_{0},A_{1},\lambda ,\eta$, and ν. This algebraic set of equations is solved by utilizing Mathematica 9. The following cases are achieved as

**Case 1:** If Ω>0,


**Set 1:**
(23)$$\begin{array}{cc} & A_{0}=B_{0}=0,k_{2}=\pm \sqrt{-k_{1}^{2}-\omega^{2}},k_{3}=\frac{3\alpha_{2}\sqrt{-k_{1}^{2}-\omega^{2}}-\alpha_{1}k_{1}-2\tau \omega }{2},a_{2}=-\frac{\sqrt{\rho^{2}-a_{1}^{2}\Omega^{2}}}{\Omega },\\ \; & \alpha_{3}=\frac{\Omega \left(\alpha_{2}^{2}\left(4\eta \nu -1\right)-\alpha_{1}^{2}+4\gamma \nu +4\eta \nu \tau^{2}-\tau^{2}\right)+2A_{1}^{2}\left(2\lambda \nu +1\right)\left(2\rho -\Omega^{2}\right)}{2\Omega }. \end{array}$$



**Set 2:**
(24)$$\begin{array}{c} A_{0}=\pm \frac{A_{1}\sqrt{2\rho -\Omega^{2}}}{\sqrt{\Omega }},k_{2}=-\sqrt{-k_{1}^{2}-\omega^{2}},k_{3}=\frac{3\alpha_{2}\sqrt{-k_{1}^{2}-\omega^{2}}-\alpha_{1}k_{1}-2\tau \omega }{2},\;a_{2}=-\frac{\sqrt{\rho^{2}-a_{1}^{2}\Omega^{2}}}{\Omega },\\ B_{0}=0,\alpha_{3}=\frac{\Omega \left(\alpha_{2}^{2}\left(4\eta \nu -1\right)-\alpha_{1}^{2}+4\gamma \nu +4\eta \nu \tau^{2}-\tau^{2}\right)+8A_{1}^{2}\left(2\lambda \nu +1\right)\left(2\rho -\Omega^{2}\right)}{2\Omega }. \end{array}$$


**Set 3:**(25)$$\begin{array}{c} A_{0}=\frac{3B_{0}\left(\frac{\Omega^{2}}{\rho }-4\right)}{2\Omega },A_{1}=0,\alpha_{3}=\frac{2\rho^{2}\Omega^{2}\left(\alpha_{2}^{2}\left(4\eta \nu -1\right)-\alpha_{1}^{2}+4\gamma \nu +\tau^{2}\left(4\eta \nu -1\right)\right)+9B_{0}^{2}\left(2\lambda \nu +1\right){\left(\Omega^{2}-4\rho \right)}^{2}}{4\rho^{2}\Omega^{2}},\\ k_{2}=\pm \sqrt{\frac{9B_{0}^{2}\left(2\lambda \nu +1\right)\left(\Omega^{2}-4\rho \right)+\rho^{2}\Omega \left(1-4\eta \nu \right)\left(k_{1}^{2}+\omega^{2}\right)}{\rho^{2}\left(\Omega -4\eta \nu \Omega \right)}},\;a_{2}=-\frac{\sqrt{4\rho^{2}\left(2\Omega^{2}-9\rho \right)-9a_{1}^{2}\Omega^{2}\left(\Omega^{2}-4\rho \right)}}{3\sqrt{\Omega^{4}-4\rho \Omega^{2}}},\\ k_{3}=\pm \frac{3\alpha_{2}\sqrt{\Omega \left(\rho^{2}\Omega \left(4\eta \nu -1\right)\left(k_{1}^{2}+\omega^{2}\right)-9B_{0}^{2}\left(2\lambda \nu +1\right)\left(4\rho -\Omega^{2}\right)\right)}}{2\sqrt{\rho^{2}\Omega^{2}\left(1-4\eta \nu \right)}}-\alpha_{1}k_{1}-2\tau \omega . \end{array}$$
Exact Solutions of Equation (Equation 1) from solutions Sets (Equation 23), (Equation 24) and (Equation 25) are constructed as
(26)$$q_{1}=-\frac{A_{1}\left(\sqrt{-\Omega }\sqrt{\rho^{2}-a_{1}^{2}\Omega^{2}}\cos\left(\xi \sqrt{-\Omega }\right)+a_{1}\Omega^{3/2}\sin\left(\xi \sqrt{\Omega }\right)\right)}{-\sqrt{\rho^{2}-a_{1}^{2}\Omega^{2}}\sin\left(\xi \sqrt{-\Omega }\right)+a_{1}\Omega \cos\left(\xi \sqrt{\Omega }\right)+\rho }e^{iP}.$$
(27)$$q_{2}=\frac{A_{1}}{\sqrt{\Omega }}\left(\pm \sqrt{2\rho -\Omega^{2}}-\frac{\lambda \Omega \left(\sqrt{\rho^{2}-a_{1}^{2}\Omega^{2}}\cos\left(\xi \sqrt{\Omega }\right)+a_{1}\Omega \sin\left(\xi \sqrt{\Omega }\right)\right)}{-\lambda \sqrt{\rho^{2}-a_{1}^{2}\Omega^{2}}\sin\left(\xi \sqrt{\Omega }\right)+a_{1}\lambda \Omega \cos\left(\xi \sqrt{\Omega }\right)+\rho \Omega }\right)e^{iP}.$$
(28)$$q_{3}=B_{0}\left(\Omega \left(\frac{1}{a_{2}\Omega \sin\left(\xi \sqrt{\Omega }\right)+a_{1}\Omega \cos\left(\xi \sqrt{\Omega }\right)+\rho }+\frac{3}{2\rho }\right)-\frac{6}{\Omega }\right)e^{iP}.$$

**Case 2:** If Ω<0,


**Set 1:**
(29)$$\begin{array}{cc} & A_{0}=B_{0}=0,k_{2}=-\sqrt{-k_{1}^{2}-\omega^{2}},k_{3}=\frac{3\alpha_{2}\sqrt{-k_{1}^{2}-\omega^{2}}-\alpha_{1}k_{1}-2\tau \omega }{2},a_{2}=\pm \frac{\sqrt{a_{1}^{2}\Omega^{2}-\rho^{2}}}{\Omega },\\ \; & \alpha_{3}=\frac{\Omega \left(\alpha_{2}^{2}\left(4\eta \nu -1\right)-\alpha_{1}^{2}+4\gamma \nu +4\eta \nu \tau^{2}-\tau^{2}\right)+2A_{1}^{2}\left(2\lambda \nu +1\right)\left(2\rho -\Omega^{2}\right)}{2\Omega }. \end{array}$$



**Set 2:**
(30)$$\begin{array}{c} A_{0}=\pm \frac{A_{1}\sqrt{2\rho -\Omega^{2}}}{\sqrt{\Omega }},k_{2}=-\sqrt{-k_{1}^{2}-\omega^{2}},k_{3}=\frac{3\alpha_{2}\sqrt{-k_{1}^{2}-\omega^{2}}-\alpha_{1}k_{1}-2\tau \omega }{2},\;a_{2}=-\frac{\sqrt{a_{1}^{2}\Omega^{2}-\rho^{2}}}{\Omega },\\ B_{0}=0,\alpha_{3}=\frac{\Omega \left(\alpha_{2}^{2}\left(4\eta \nu -1\right)-\alpha_{1}^{2}+4\gamma \nu +4\eta \nu \tau^{2}-\tau^{2}\right)+8A_{1}^{2}\left(2\lambda \nu +1\right)\left(2\rho -\Omega^{2}\right)}{2\Omega }. \end{array}$$


**Set 3:**(31)$$\begin{array}{c} A_{0}=\frac{3B_{0}\left(\frac{\Omega^{2}}{\rho }-4\right)}{2\Omega },A_{1}=0,\alpha_{3}=\frac{2\rho^{2}\Omega^{2}\left(\alpha_{2}^{2}\left(4\eta \nu -1\right)-\alpha_{1}^{2}+4\gamma \nu +4\eta \nu \tau^{2}-\tau^{2}\right)+9B_{0}^{2}\left(2\lambda \nu +1\right){\left(\Omega^{2}-4\rho \right)}^{2}}{4\rho^{2}\Omega^{2}},\\ k_{2}=\mp \sqrt{\frac{\Omega \left(\rho^{2}\Omega \left(4\eta \nu -1\right)\left(k_{1}^{2}+\omega^{2}\right)-9B_{0}^{2}\left(2\lambda \nu +1\right)\left(4\rho -\Omega^{2}\right)\right)}{\rho^{2}\Omega^{2}\left(1-4\eta \nu \right)}},a_{2}=-\frac{\sqrt{9a_{1}^{2}\left(4\rho \Omega^{2}-\Omega^{4}\right)+4\rho^{2}\left(2\Omega^{2}-9\rho \right)}}{3\sqrt{4\rho \Omega^{2}-\Omega^{4}}},\\ k_{3}=\mp \frac{3\alpha_{2}\sqrt{\rho^{2}\Omega \left(4\eta \nu -1\right)\left(k_{1}^{2}+\omega^{2}\right)-9B_{0}^{2}\left(2\lambda \nu +1\right)\left(4\rho -\Omega^{2}\right)}}{2\sqrt{\rho^{2}\left(1-4\eta \nu \right)}}-\alpha_{1}k_{1}-2\tau \omega . \end{array}$$
Solitons solutions of Equation (Equation 1) from solutions (Equation 29), (Equation 30), and (Equation 31) as:(32)$$q_{4}=\frac{A_{1}\lambda \sqrt{-\Omega }\left(a_{1}\Omega \sinh\left(\xi \sqrt{-\Omega }\right)-\sqrt{a_{1}^{2}\Omega^{2}-\rho^{2}}\cosh\left(\xi \sqrt{-\Omega }\right)\right)}{-\lambda \sqrt{a_{1}^{2}\Omega^{2}-\rho^{2}}\sinh\left(\xi \sqrt{-\Omega }\right)+a_{1}\lambda \Omega \cosh\left(\xi \sqrt{-\Omega }\right)+\rho \Omega }e^{iP}.$$
(33)$$q_{5}=A_{1}\left(\frac{\lambda \sqrt{-\Omega }\left(a_{1}\Omega \sinh\left(\xi \sqrt{-\Omega }\right)-\sqrt{a_{1}^{2}\Omega^{2}-\rho^{2}}\cosh\left(\xi \sqrt{-\Omega }\right)\right)}{-\lambda \sqrt{a_{1}^{2}\Omega^{2}-\rho^{2}}\sinh\left(\xi \sqrt{-\Omega }\right)+a_{1}\lambda \Omega \cosh\left(\xi \sqrt{-\Omega }\right)+\rho \Omega }\pm \frac{\sqrt{2\rho -\Omega^{2}}}{\sqrt{\Omega }}\right)e^{iP}.$$
(34)$$q_{6}=B_{0}\left(\frac{\lambda }{a_{2}\lambda \sinh\left(\xi \sqrt{-\Omega }\right)+a_{1}\lambda \cosh\left(\xi \sqrt{-\Omega }\right)+\rho }+\frac{3\Omega }{2\rho }-\frac{6}{\Omega }\right)e^{iP}.$$

**Case 3:** If Ω=0,


**Set 1:**
(35)$$\begin{array}{c} A_{0}=0,B_{0}=\pm \sqrt{A_{1}^{2}\left(a_{1}^{2}-2a_{2}^{2}\rho \right)},k_{3}=\frac{1}{2}\left(\frac{3\alpha_{2}\sqrt{A_{1}^{2}\left(8\lambda \nu +4\right)-\left(4\eta \nu -1\right)\left(k_{1}^{2}+\omega^{2}\right)}}{\sqrt{4\eta \nu -1}}-\alpha_{1}k_{1}-2\tau \omega \right),\\ \alpha_{3}=\frac{\alpha_{2}^{2}\left(4\eta \nu -1\right)-\alpha_{1}^{2}+4\gamma \nu +\tau^{2}\left(4\eta \nu -1\right)}{2},\;k_{2}=-\frac{\sqrt{A_{1}^{2}\left(8\lambda \nu +4\right)-\left(4\eta \nu -1\right)\left(k_{1}^{2}+\omega^{2}\right)}}{\sqrt{4\eta \nu -1}}. \end{array}$$



**Set 2:**
(36)$$\begin{array}{cc} & A_{0}=B_{0}=0,a_{1}=\sqrt{2}a_{2}\sqrt{\rho },\alpha_{3}=\frac{1}{2}\left(\alpha_{2}^{2}\left(4\eta \nu -1\right)-\alpha_{1}^{2}+4\gamma \nu +\tau^{2}\left(4\eta \nu -1\right)\right),\\ & k_{2}=\pm \frac{\sqrt{A_{1}^{2}\left(8\lambda \nu +4\right)-\left(4\eta \nu -1\right)\left(k_{1}^{2}+\omega^{2}\right)}}{\sqrt{4\eta \nu -1}},\\ \; & k_{3}=\frac{1}{2}\left(\mp \frac{3\alpha_{2}\sqrt{A_{1}^{2}\left(8\lambda \nu +4\right)-\left(4\eta \nu -1\right)\left(k_{1}^{2}+\omega^{2}\right)}}{\sqrt{4\eta \nu -1}}-\alpha_{1}k_{1}-2\tau \omega \right). \end{array}$$


**Set 3:**(37)$$\begin{array}{c} A_{0}=B_{0}=0,\alpha_{3}=\frac{1}{2},\alpha_{2}^{2}\left(4\eta \nu -1\right)-\alpha_{1}^{2}+4\gamma \nu +\tau^{2}\left(4\eta \nu -1\right),k_{2}=\mp \frac{\sqrt{A_{1}^{2}\left(8\lambda \nu +4\right)-\left(4\eta \nu -1\right)\left(k_{1}^{2}+\omega^{2}\right)}}{\sqrt{4\eta \nu -1}},\\ k_{3}=\frac{1}{2}\left(\pm \frac{3\alpha_{2}\sqrt{A_{1}^{2}\left(8\lambda \nu +4\right)-\left(4\eta \nu -1\right)\left(k_{1}^{2}+\omega^{2}\right)}}{\sqrt{4\eta \nu -1}}-\alpha_{1}k_{1}-2\tau \omega \right),a_{1}=-\sqrt{2}a_{2}\sqrt{\rho }. \end{array}$$
Solitons solutions of Equation (Equation 1) from solutions (Equation 35), (Equation 36) and (Equation 37) are constructed as
(38)$$q_{7}\left(\xi \right)=\frac{2\left(A_{1}\left(a_{1}+\xi \rho \right)\pm \sqrt{A_{1}^{2}\left(a_{1}^{2}-2a_{2}^{2}\rho \right)}\right)}{2a_{1}\xi +2a_{2}+\xi^{2}\rho }e^{iP}.$$
(39)$$q_{8}\left(\xi \right)=\frac{2A_{1}\left(\sqrt{2}a_{2}\sqrt{\rho }+\xi \rho \right)}{2a_{2}\left(\sqrt{2}\xi \sqrt{\rho }+1\right)+\xi^{2}\rho }e^{iP}.$$
(40)$$q_{9}=\frac{2A_{1}\left(\xi \rho -\sqrt{2}a_{2}\sqrt{\rho }\right)}{a_{2}\left(2-2\sqrt{2}\xi \sqrt{\rho }\right)+\xi^{2}\rho }e^{iP}.$$

### 4.2. Solitons Solutions by the Exp(−ϕ(ξ))-Expansion Method

In this sub-part, the exp(ϕ(ξ))-expansion method is employed to construct the solution of Equation (Equation 1). Apply the homogeneous balance principle on Equation (Equation 21) and assume the solution as
(41)$$\psi \left(\xi \right)=\frac{B_{-1}}{\exp(-\phi (\xi ))}+B_{0}+B_{1}\exp\left(-\phi \left(\xi \right)\right).$$
Substituting Equation (Equation 41) along Equation (Equation 10) in Equation (Equation 21) and setting the coefficients to zero of ${\left(\exp(-\phi (x))\right)}^{i}$, we obtain a set of equations in parameters $B_{-1},\;B_{0},B_{1},\lambda ,\eta ,a,b$ and ν. This algebraic set of equations are solved by utilizing Mathematica 9. The following sets are achieved as


**Set 1:**
(42)$$\begin{array}{cc} & B_{-1}=0,B_{0}=\frac{bB_{1}}{2},k_{3}=\frac{-\alpha_{1}k_{1}-3\alpha_{2}k_{2}-2\tau \omega }{2},\;\lambda =\frac{\left(4\eta \nu -1\right)\left(k_{1}^{2}+k_{2}^{2}+\omega^{2}\right)-B_{1}^{2}}{2B_{1}^{2}\nu },\\ & \alpha_{3}=\frac{1}{4}\left(-k_{1}^{2}\left(4a-b^{2}\right)\left(4\eta \nu -1\right)-k_{2}^{2}\left(4a-b^{2}\right)\left(4\eta \nu -1\right)-16a\eta \nu \omega^{2}+8\alpha_{2}^{2}\eta \nu +4a\omega^{2}\right.\\ \; & \;\;\;\;\;\;\;\;\left.-2\alpha_{1}^{2}-2\alpha_{2}^{2}+4b^{2}\eta \nu \omega^{2}-b^{2}\omega^{2}+8\gamma \nu +8\eta \nu \tau^{2}-2\tau^{2}\right). \end{array}$$



**Set 2:**
(43)$$\begin{array}{cc} & B_{0}=\frac{bB_{-1}}{2a},B_{1}=0,k_{3}=\frac{-\alpha_{1}k_{1}-3\alpha_{2}k_{2}-2\tau \omega }{2},\lambda =\frac{a^{2}\left(4\eta \nu -1\right)\left(k_{1}^{2}+k_{2}^{2}+\omega^{2}\right)-B_{-1}^{2}}{2B_{-1}^{2}\nu },\\ & \alpha_{3}=\frac{1}{4}\left(-k_{1}^{2}\left(4a-b^{2}\right)\left(4\eta \nu -1\right)-k_{2}^{2}\left(4a-b^{2}\right)\left(4\eta \nu -1\right)-16a\eta \nu \omega^{2}+8\alpha_{2}^{2}\eta \nu +4a\omega^{2}-2\alpha_{1}^{2}\right.\\ \; & \;\;\;\;\;\;\;\;\;\;\left.-2\alpha_{2}^{2}+4b^{2}\eta \nu \omega^{2}-b^{2}\omega^{2}+8\gamma \nu +8\eta \nu \tau^{2}-2\tau^{2}\right). \end{array}$$



**Set 3:**
(44)$$\begin{array}{cc} & k_{3}=-\frac{\alpha_{1}k_{1}}{2}-\frac{3\alpha_{2}k_{2}}{2}+\sqrt{-k_{1}^{2}-k_{2}^{2}}\tau ,\alpha_{3}=\frac{\alpha_{2}^{2}\left(4\eta \nu -1\right)-\alpha_{1}^{2}+4\gamma \nu +4\eta \nu \tau^{2}-\tau^{2}}{2},\;\\ \; & \lambda =-\frac{1}{2\nu },\;\omega =-\sqrt{-k_{1}^{2}-k_{2}^{2}}. \end{array}$$



**Set 4:**
(45)$$\begin{array}{cc} & B_{-1}=0,k_{3}=-\frac{\alpha_{1}k_{1}}{2}-\frac{3\alpha_{2}k_{2}}{2}+\sqrt{-k_{1}^{2}-k_{2}^{2}}\tau ,\alpha_{3}=\frac{\alpha_{2}^{2}\left(4\eta \nu -1\right)-\alpha_{1}^{2}+4\gamma \nu +4\eta \nu \tau^{2}-\tau^{2}}{2},\\ \; & \lambda =-\frac{1}{2\nu },\omega =\pm \sqrt{-k_{1}^{2}-k_{2}^{2}}. \end{array}$$



**Set 5:**
(46)$$\begin{array}{cc} & B_{1}=0,k_{3}=-\frac{\alpha_{1}k_{1}}{2}-\frac{3\alpha_{2}k_{2}}{2}+\sqrt{-k_{1}^{2}-k_{2}^{2}}\tau ,\alpha_{3}=\frac{\alpha_{2}^{2}\left(4\eta \nu -1\right)-\alpha_{1}^{2}+4\gamma \nu +4\eta \nu \tau^{2}-\tau^{2}}{2},\\ \; & \lambda =-\frac{1}{2\nu },\omega =-\sqrt{-k_{1}^{2}-k_{2}^{2}}. \end{array}$$



**Set 6:**
(47)$$\begin{array}{cc} & B_{-1}=\frac{2aB_{0}}{b},B_{1}=0,\alpha_{1}=-\frac{3\alpha_{2}k_{2}+2k_{3}+2\tau \omega }{k_{1}},\lambda =\frac{b^{2}\left(4\eta \nu -1\right)\left(k_{1}^{2}+k_{2}^{2}+\omega^{2}\right)-4B_{0}^{2}}{8B_{0}^{2}\nu },\\ & \alpha_{3}=\left(k_{1}^{2}\left(-k_{2}^{2}\left(4a-b^{2}\right)\left(4\eta \nu -1\right)-16a\eta \nu \omega^{2}+\alpha_{2}^{2}\left(8\eta \nu -2\right)+4a\omega^{2}+4b^{2}\eta \nu \omega^{2}-b^{2}\omega^{2}+8\gamma \nu \right.\right.\\ \; & \;\;\;\;\;\;\;\left.\left.+8\eta \nu \tau^{2}-2\tau^{2}\right)+k_{1}^{4}\left(-\left(4a-b^{2}\right)\right)\left(4\eta \nu -1\right)-2{\left(3\alpha_{2}k_{2}+2k_{3}+2\tau \omega \right)}^{2}\right)/4k_{1}^{2}. \end{array}$$


**Set 7:**(48)$$\begin{array}{cc} & B_{-1}=0,B_{0}=\frac{bB_{1}}{2},\alpha_{1}=-\frac{3\alpha_{2}k_{2}+2k_{3}+2\tau \omega }{k_{1}},\lambda =\frac{\left(4\eta \nu -1\right)\left(k_{1}^{2}+k_{2}^{2}+\omega^{2}\right)-B_{1}^{2}}{2B_{1}^{2}\nu },\\ & \alpha_{3}=\left(k_{1}^{2}\left(-k_{2}^{2}\left(4a-b^{2}\right)\left(4\eta \nu -1\right)-16a\eta \nu \omega^{2}+\alpha_{2}^{2}\left(8\eta \nu -2\right)+4a\omega^{2}+4b^{2}\eta \nu \omega^{2}-b^{2}\omega^{2}\right.\right.\\ \; & \;\;\;\;\;\left.\left.+8\gamma \nu +8\eta \nu \tau^{2}-2\tau^{2}\right)+k_{1}^{4}\left(-\left(4a-b^{2}\right)\right)\left(4\eta \nu -1\right)-2{\left(3\alpha_{2}k_{2}+2k_{3}+2\tau \omega \right)}^{2}\right)/4k_{1}^{2}. \end{array}$$
From set 1, the following five types of solutions of Equation (Equation 1) are constructed as 

**Type I:** When $a\neq 0,b^{2}-4a>0$,
(49)$$q_{1}\left(\xi \right)=\frac{B_{1}}{2}\left(b-\frac{4a}{\sqrt{b^{2}-4a}\tanh\left(\frac{\sqrt{b^{2}-4a}}{2}\left(\xi +c\right)\right)+b}\right)e^{iP}.$$

**Type II** When $a\neq 0,b^{2}-4a<0$,
(50)$$q_{2}\left(\xi \right)=\frac{B_{1}}{2}\left(b-\frac{4a}{b-\sqrt{4a-b^{2}}\tan\left(\frac{\sqrt{4a-b^{2}}}{2}\left(\xi +c\right)\right)}\right)e^{iP}.$$

**Type III** When $a=0,b\neq 0\;\&\;b^{2}-4a>0$,
(51)$$q_{3}\left(\xi \right)=\frac{bB_{1}}{2}\left(\frac{2}{e^{b(\xi +c)}-1}+1\right)e^{iP}.\;\;\;\;\;\;$$

**Type IV** When $a\neq ,b\neq 0\;\&\;b^{2}-4a=0$,
(52)$$q_{4}\left(\xi \right)=\frac{bB_{1}}{2b\left(\xi +c\right)+2}e^{iP}.\;\;\;\;\;\;\;\;\;\;\;\;\;$$

**Type V** When $a=0,=0\;\&\;b^{2}-4a=0$,
(53)$$q_{5}\left(\xi \right)=\frac{B_{1}}{2}\left(b+\frac{2}{\xi +c}\right)e^{iP}.$$
Similarly, further novel exact solutions of (Equation 1) forming other solution sets can be constructed.

### 4.3. Solitons Solutions by the Proposed F-Expansion Method

In this sub-part, the proposed F-Expansion technique is employed to construct the solitons and wave solutions of Equation (Equation 1). Apply the homogeneous balance principle on Equation (Equation 21) and assume the solution of Equation (Equation 21) as
(54)$$\psi \left(\xi \right)=A_{0}+A_{1}\left(a+F(\xi )\right)+\frac{B_{1}}{a+F(\xi )}.$$
Substituting Equation (Equation 54) along Equation (Equation 17) in Equation (Equation 21) and setting the coefficients to zero of $\frac{F^{j}\left(\xi \right)}{{\left(a+F(\xi )\right)}^{k}}$, we obtain a set of equations in parameters $A_{0},A_{1},B_{1},d_{0},d_{1},d_{2},d_{3}$, and ν. This algebraic set of equations are solved by utilizing Mathematica 9. The following cases of solutions are achieved as

**Case 1:** If $d_{0}=d_{2}=0$,
(55)$$\begin{array}{cc} & A_{0}=-aA_{1},\;B_{1}=0,\;\alpha_{3}=\frac{d_{3}\left(\gamma -\alpha_{1}^{2}\eta \right)-A_{1}^{2}d_{1}\left(2\eta +\lambda \right)}{2d_{3}\eta },\;\nu =\frac{1}{4\eta },\\ \; & \;k_{3}=-\frac{A_{1}^{2}\left(2\eta +\lambda \right)+4d_{3}\eta^{2}\left(\alpha_{1}k_{1}+3\alpha_{2}k_{2}+2\tau \omega \right)}{8d_{3}\eta^{2}}. \end{array}$$
(56)$$\begin{array}{cc} & A_{0}=B_{1}=a=0,\;k_{3}=-\frac{A_{1}^{2}\left(2\eta +\lambda \right)+4d_{3}\eta^{2}\left(\alpha_{1}k_{1}+3\alpha_{2}k_{2}+2\tau \omega \right)}{8d_{3}\eta^{2}},\\ \; & \;\nu =\frac{1}{4\eta },\alpha_{3}=\frac{d_{3}\left(\gamma -\alpha_{1}^{2}\eta \right)-A_{1}^{2}d_{1}\left(2\eta +\lambda \right)}{2d_{3}\eta }. \end{array}$$
The solitons solutions of Equation (Equation 1) from solutions (Equation 55) and (Equation 56) are constructed as
(57)$$q_{11}\left(\xi \right)=\frac{A_{1}\sqrt{d_{1}}e^{d_{1}\xi }}{\sqrt{1-d_{3}e^{2d_{1}\xi }}}e^{i\left(\alpha_{1}x+\alpha_{2}y+\alpha_{3}z+t\tau +\epsilon \right)},\;\;\;\;\;d_{1}>0.$$
(58)$$q_{12}\left(\xi \right)=\frac{A_{1}\sqrt{-d_{1}}}{\sqrt{e^{-2d_{1}\xi }+d_{3}}}e^{i\left(\alpha_{1}x+\alpha_{2}y+\alpha_{3}z+t\tau +\epsilon \right)},\;\;\;\;\;d_{1}<0.$$
(59)$$q_{13}\left(\xi \right)=\frac{A_{1}\sqrt{d_{1}}e^{d_{1}\xi }}{\sqrt{1-d_{3}e^{2d_{1}\xi }}}e^{i\left(\alpha_{1}x+\alpha_{2}y+\alpha_{3}z+t\tau +\epsilon \right)},\;\;\;\;\;d_{1}>0.$$
(60)$$q_{14}\left(\xi \right)=\frac{A_{1}\sqrt{-d_{1}}}{\sqrt{e^{-2d_{1}\xi }+d_{3}}}e^{i\left(\alpha_{1}x+\alpha_{2}y+\alpha_{3}z+t\tau +\epsilon \right)},\;\;\;\;\;d_{1}<0.$$

**Case 2:** If $d_{0}=d_{3}=0$,
(61)$$\begin{array}{cc} & A_{0}=0,\;B_{1}=0,\;\alpha_{3}=\frac{A_{1}^{2}d_{1}^{2}\left(2\lambda \nu +1\right)+2d_{2}^{2}\left(\alpha_{2}^{2}\left(4\eta \nu -1\right)-\alpha_{1}^{2}+4\gamma \nu +\tau^{2}\left(4\eta \nu -1\right)\right)}{4d_{2}^{2}},\\ & \;a=\frac{d_{1}}{2d_{2}},\;\;k_{3}=\frac{1}{2}\left(\frac{3\alpha_{2}\sqrt{A_{1}^{2}\left(2\lambda \nu +1\right)-d_{2}^{2}\left(4\eta \nu -1\right)\left(k_{1}^{2}+\omega^{2}\right)}}{\sqrt{d_{2}^{2}\left(4\eta \nu -1\right)}}-\alpha_{1}k_{1}-2\tau \omega \right),\\ \; & \;k_{2}=-\frac{\sqrt{A_{1}^{2}\left(2\lambda \nu +1\right)-d_{2}^{2}\left(4\eta \nu -1\right)\left(k_{1}^{2}+\omega^{2}\right)}}{\sqrt{d_{2}^{2}\left(4\eta \nu -1\right)}}. \end{array}$$
(62)$$\begin{array}{cc} & A_{0}=0,\;B_{1}=-\frac{A_{1}d_{1}^{2}}{4d_{2}^{2}},\;\alpha_{3}=\frac{d_{2}^{2}\left(\alpha_{2}^{2}\left(4\eta \nu -1\right)-\alpha_{1}^{2}+4\gamma \nu +\tau^{2}\left(4\eta \nu -1\right)\right)-A_{1}^{2}d_{1}^{2}\left(2\lambda \nu +1\right)}{2d_{2}^{2}},\\ & \;a=\frac{d_{1}}{2d_{2}},\;k_{3}=\frac{1}{2}\left(\frac{3\alpha_{2}\sqrt{A_{1}^{2}\left(2\lambda \nu +1\right)-d_{2}^{2}\left(4\eta \nu -1\right)\left(k_{1}^{2}+\omega^{2}\right)}}{\sqrt{d_{2}^{2}\left(4\eta \nu -1\right)}}-\alpha_{1}k_{1}-2\tau \omega \right),\\ \; & \;k_{2}=-\frac{\sqrt{A_{1}^{2}\left(2\lambda \nu +1\right)-d_{2}^{2}\left(4\eta \nu -1\right)\left(k_{1}^{2}+\omega^{2}\right)}}{\sqrt{d_{2}^{2}\left(4\eta \nu -1\right)}}. \end{array}$$
(63)$$\begin{array}{cc} & A_{0}=\frac{A_{1}}{2}\left(\frac{d_{1}}{d_{2}}-2a\right),\;\alpha_{3}=\frac{A_{1}^{2}d_{1}^{2}\left(2\lambda \nu +1\right)+2d_{2}^{2}\left(\alpha_{2}^{2}\left(4\eta \nu -1\right)-\alpha_{1}^{2}+4\gamma \nu +\tau^{2}\left(4\eta \nu -1\right)\right)}{4d_{2}^{2}},\\ & \;B_{1}=0,\;k_{3}=\frac{1}{2}\left(\frac{3\alpha_{2}\sqrt{A_{1}^{2}\left(2\lambda \nu +1\right)-d_{2}^{2}\left(4\eta \nu -1\right)\left(k_{1}^{2}+\omega^{2}\right)}}{\sqrt{d_{2}^{2}\left(4\eta \nu -1\right)}}-\alpha_{1}k_{1}-2\tau \omega \right),\\ \; & \;k_{2}=-\frac{\sqrt{A_{1}^{2}\left(2\lambda \nu +1\right)-d_{2}^{2}\left(4\eta \nu -1\right)\left(k_{1}^{2}+\omega^{2}\right)}}{\sqrt{d_{2}^{2}\left(4\eta \nu -1\right)}}. \end{array}$$
The solitons of Equation (Equation 1) from solutions (Equation 61) and (Equation 62) are constructed as
(64)$$q_{21}\left(\xi \right)=\frac{A_{1}d_{1}\left(d_{2}e^{d_{1}\xi }+1\right)}{2d_{2}\left(1-d_{2}e^{d_{1}\xi }\right)}e^{i\left(\alpha_{1}x+\alpha_{2}y+\alpha_{3}z+t\tau +\epsilon \right)},\;\;\;\;\;d_{1}>0.$$
(65)$$q_{22}\left(\xi \right)=\frac{A_{1}d_{1}\left(1-d_{2}e^{d_{1}\xi }\right)}{2d_{2}\left(d_{2}e^{d_{1}\xi }+1\right)}e^{i\left(\alpha_{1}x+\alpha_{2}y+\alpha_{3}z+t\tau +\epsilon \right)},\;\;\;\;\;d_{1}<0.$$
(66)$$q_{23}\left(\xi \right)=\frac{2A_{1}d_{1}e^{d_{1}\xi }}{1-d_{2}^{2}e^{2d_{1}\xi }}e^{i\left(\alpha_{1}x+\alpha_{2}y+\alpha_{3}z+t\tau +\epsilon \right)},\;\;\;\;\;d_{1}>0.$$
(67)$$q_{24}\left(\xi \right)=\frac{2A_{1}d_{1}e^{d_{1}\xi }}{d_{2}^{2}e^{2d_{1}\xi }-1}e^{i\left(\alpha_{1}x+\alpha_{2}y+\alpha_{3}z+t\tau +\epsilon \right)},\;\;\;\;\;d_{1}<0.$$
Similarly, more generalized results can be constructed of Equation (Equation 1) from solution (Equation 63).

**Case 3:** If $d_{1}=d_{3}=0$,
(68)$$\begin{array}{cc} & A_{0}=-aA_{1},\;\alpha_{3}=\frac{d_{2}\left(\alpha_{2}^{2}\left(4\eta \nu -1\right)-\alpha_{1}^{2}+4\gamma \nu +\tau^{2}\left(4\eta \nu -1\right)\right)-2A_{1}^{2}d_{0}\left(2\lambda \nu +1\right)}{2d_{2}},\\ & \;B_{1}=0,\;k_{3}=-\frac{1}{2}\left(\frac{3\alpha_{2}\sqrt{A_{1}^{2}\left(2\lambda \nu +1\right)-d_{2}^{2}\left(4\eta \nu -1\right)\left(k_{1}^{2}+\omega^{2}\right)}}{\sqrt{d_{2}^{2}\left(4\eta \nu -1\right)}}+\alpha_{1}k_{1}+2\tau \omega \right),\\ \; & \;k_{2}=\frac{\sqrt{A_{1}^{2}\left(2\lambda \nu +1\right)-d_{2}^{2}\left(4\eta \nu -1\right)\left(k_{1}^{2}+\omega^{2}\right)}}{\sqrt{d_{2}^{2}\left(4\eta \nu -1\right)}}. \end{array}$$
(69)$$\begin{array}{cc} & A_{0}=0,\;a=0,\;\alpha_{3}=\frac{d_{2}\left(\alpha_{2}^{2}\left(4\eta \nu -1\right)-\alpha_{1}^{2}+4\gamma \nu +\tau^{2}\left(4\eta \nu -1\right)\right)-8A_{1}^{2}d_{0}\left(2\lambda \nu +1\right)}{2d_{2}},\\ & \;B_{1}=-\frac{A_{1}d_{0}}{d_{2}},\;k_{3}=\frac{1}{2}\left(\frac{3\alpha_{2}\sqrt{A_{1}^{2}\left(2\lambda \nu +1\right)-d_{2}^{2}\left(4\eta \nu -1\right)\left(k_{1}^{2}+\omega^{2}\right)}}{\sqrt{d_{2}^{2}\left(4\eta \nu -1\right)}}-\alpha_{1}k_{1}-2\tau \omega \right),\\ \; & \;k_{2}=-\frac{\sqrt{A_{1}^{2}\left(2\lambda \nu +1\right)-d_{2}^{2}\left(4\eta \nu -1\right)\left(k_{1}^{2}+\omega^{2}\right)}}{\sqrt{d_{2}^{2}\left(4\eta \nu -1\right)}}. \end{array}$$
(70)$$\begin{array}{cc} & A_{0}=0,\;\alpha_{3}=\frac{d_{2}\left(\alpha_{2}^{2}\left(4\eta \nu -1\right)-\alpha_{1}^{2}+4\gamma \nu +\tau^{2}\left(4\eta \nu -1\right)\right)-2A_{1}^{2}d_{0}\left(2\lambda \nu +1\right)}{2d_{2}},\\ & \;B_{1}=0,\;a=0,\;k_{2}=-\frac{\sqrt{A_{1}^{2}\left(2\lambda \nu +1\right)-d_{2}^{2}\left(4\eta \nu -1\right)\left(k_{1}^{2}+\omega^{2}\right)}}{\sqrt{d_{2}^{2}\left(4\eta \nu -1\right)}},\\ \; & \;k_{3}=\frac{1}{2}\left(\frac{3\alpha_{2}\sqrt{A_{1}^{2}\left(2\lambda \nu +1\right)-d_{2}^{2}\left(4\eta \nu -1\right)\left(k_{1}^{2}+\omega^{2}\right)}}{\sqrt{d_{2}^{2}\left(4\eta \nu -1\right)}}-\alpha_{1}k_{1}-2\tau \omega \right). \end{array}$$
The solitons solutions of Equation (Equation 1) from solutions (Equation 68) and (Equation 69) are constructed as
(71)$$q_{31}\left(\xi \right)=\frac{A_{1}d_{0}\tan\left(\sqrt{d_{0}d_{2}}\xi \right)}{\sqrt{d_{0}d_{2}}}e^{i\left(\alpha_{1}x+\alpha_{2}y+\alpha_{3}z+t\tau +\epsilon \right)},\;\;\;\;\;\;\;\;\;\;d_{0}d_{2}>0.$$
(72)$$q_{32}\left(\xi \right)=\frac{A_{1}d_{0}\tanh\left(\sqrt{-d_{0}d_{2}}\xi \right)}{\sqrt{-d_{0}d_{2}}}e^{i\left(\alpha_{1}x+\alpha_{2}y+\alpha_{3}z+t\tau +\epsilon \right)},\;\;\;\;\;d_{0}d_{2}<0.$$
(73)$$q_{33}\left(\xi \right)=\frac{A_{1}d_{0}\tan\left(\sqrt{d_{0}d_{2}}\xi \right)\left(1-\cot^{2}\left(\sqrt{d_{0}d_{2}}\xi \right)\right)}{\sqrt{d_{0}d_{2}}}e^{i\left(\alpha_{1}x+\alpha_{2}y+\alpha_{3}z+t\tau +\epsilon \right)},\;\;\;\;\;\;\;\;\;\;\;\;\;\;\;\;d_{0}d_{2}>0.$$
(74)$$q_{34}\left(\xi \right)=\frac{A_{1}d_{0}\tanh\left(\sqrt{-d_{0}d_{2}}\xi \right)\left(\coth^{2}\left(\sqrt{-d_{0}d_{2}}\xi \right)+1\right)}{\sqrt{-d_{0}d_{2}}}e^{i\left(\alpha_{1}x+\alpha_{2}y+\alpha_{3}z+t\tau +\epsilon \right)},\;\;\;\;\;d_{0}d_{2}<0.$$
Similarly, more results can be constructed of Equation (Equation 1) from solutions (Equation 63).

**Case 4:** If $d_{3}=0$,
(75)$$\begin{array}{c} A_{0}=\frac{B_{1}\left(d_{1}-2ad_{2}\right)}{2\left(a\left(ad_{2}-d_{1}\right)+d_{0}\right)},\;k_{2}=-\frac{\sqrt{B_{1}^{2}\left(2\lambda \nu +1\right)-{\left(a\left(ad_{2}-d_{1}\right)+d_{0}\right)}^{2}\left(4\eta \nu -1\right)\left(k_{1}^{2}+\omega^{2}\right)}}{\sqrt{{\left(a\left(ad_{2}-d_{1}\right)+d_{0}\right)}^{2}\left(4\eta \nu -1\right)}},\\ \;A_{1}=0,\;k_{3}=\frac{1}{2}\left(\frac{3\alpha_{2}\sqrt{B_{1}^{2}\left(2\lambda \nu +1\right)-{\left(a^{2}d_{2}-ad_{1}+d_{0}\right)}^{2}\left(4\eta \nu -1\right)\left(k_{1}^{2}+\omega^{2}\right)}}{\sqrt{{\left(a\left(ad_{2}-d_{1}\right)+d_{0}\right)}^{2}\left(4\eta \nu -1\right)}}-\alpha_{1}k_{1}-2\tau \omega \right),\\ \;\alpha_{3}=\left(2a^{4}d_{2}^{2}\left(\alpha_{2}^{2}\left(4\eta \nu -1\right)-\alpha_{1}^{2}+4\gamma \nu +\tau^{2}\left(4\eta \nu -1\right)\right)+4a^{3}d_{1}d_{2}\left(\alpha_{2}^{2}\left(1-4\eta \nu \right)+\alpha_{1}^{2}-4\nu \left(\gamma +\eta \tau^{2}\right)\right.\right.\\ \;\;\;\;\;\;\left.+\tau^{2}\right)+d_{1}^{2}\left(2a^{2}\left(\alpha_{2}^{2}\left(4\eta \nu -1\right)-\alpha_{1}^{2}+4\gamma \nu +\tau^{2}\left(4\eta \nu -1\right)\right)+B_{1}^{2}\left(2\lambda \nu +1\right)\right)+4d_{0}\left(d_{2}\left(a^{2}\left(\alpha_{2}^{2}(4\eta \nu \right.\right.\right.\\ \;\;\;\;\;\;\left.\left.\left.-1)-\alpha_{1}^{2}+4\gamma \nu +\tau^{2}\left(4\eta \nu -1\right)\right)-B_{1}^{2}\left(2\lambda \nu +1\right)\right)+ad_{1}\left(\alpha_{2}^{2}\left(1-4\eta \nu \right)+\alpha_{1}^{2}-4\nu \left(\gamma +\eta \tau^{2}\right)\right.\right.\\ \;\;\;\;\;\;\;\left.\left.\left.+\tau^{2}\right)\right)+d_{0}^{2}\left(\alpha_{2}^{2}\left(8\eta \nu -2\right)-2\alpha_{1}^{2}+8\gamma \nu +2\tau^{2}\left(4\eta \nu -1\right)\right)\right)/\left(4{\left(a\left(ad_{2}-d_{1}\right)+d_{0}\right)}^{2}\right). \end{array}$$
(76)$$\begin{array}{c} A_{0}=\frac{A_{1}\left(d_{1}-2ad_{2}\right)-\sqrt{A_{1}^{2}\left(d_{1}^{2}-4d_{0}d_{2}\right)}}{2d_{2}},\;k_{2}=-\frac{\sqrt{A_{1}^{2}\left(2\lambda \nu +1\right)-d_{2}^{2}\left(4\eta \nu -1\right)\left(k_{1}^{2}+\omega^{2}\right)}}{\sqrt{d_{2}^{2}\left(4\eta \nu -1\right)}},\\ B_{1}=0,\;\alpha_{3}=\frac{2A_{1}^{2}\left(d_{1}^{2}-4d_{0}d_{2}\right)\left(2\lambda \nu +1\right)+d_{2}^{2}\left(\alpha_{2}^{2}\left(4\eta \nu -1\right)-\alpha_{1}^{2}+4\gamma \nu +\tau^{2}\left(4\eta \nu -1\right)\right)}{2d_{2}^{2}},\\ k_{3}=\left(3\left(4\alpha_{2}\eta d_{2}\sqrt{\left(4\eta \nu -1\right)\left(A_{1}^{2}\left(2\lambda \nu +1\right)-d_{2}^{2}\left(4\eta \nu -1\right)\left(k_{1}^{2}+\omega^{2}\right)\right)}+A_{1}^{2}\sqrt{d_{1}^{2}-4d_{0}d_{2}}(4\eta \nu \right.\right.\\ \;\;\;\;\;\;\;\left.\left.-1)(2\lambda \nu +1)\right)-4d_{2}^{2}\eta \left(4\eta \nu -1\right)\left(\alpha_{1}k_{1}+2\tau \omega \right)\right)/\left(8d_{2}^{2}\eta \left(4\eta \nu -1\right)\right). \end{array}$$
The solitons solutions of Equation (Equation 1) from solutions (Equation 75) and (Equation 76) are constructed as
(77)$$\begin{array}{cc} q_{41}\left(\xi \right)= & \frac{B_{1}e^{i\left(\alpha_{1}x+\alpha_{2}y+\alpha_{3}z+t\tau +\epsilon \right)}}{2}\left(\frac{4d_{2}}{2ad_{2}+\sqrt{4d_{0}d_{2}-d_{1}^{2}}\tan\left(\frac{\sqrt{4d_{0}d_{2}-d_{1}^{2}}}{2}\xi \right)-d_{1}}\right.\\ \; & \;\;\;\;\;\;\;\;\;\;\;\;\;\;\;\;\;\;\;\;\;\;\;\;\;\;\;\;\;\;\;\;\;\;\;\;\;\;\;\;\;\;\;\;\;\;\;\;\;\;\;\;\;\;\;\;\;\;\;\;\;\;\;\;\;\;\;\;\;\;\;\;\;\;\;\;\;\;\;\;\;\;\;\;\left.+\frac{d_{1}-2ad_{2}}{a^{2}d_{2}-ad_{1}+d_{0}}\right),\;\;4d_{0}d_{2}>d_{1}^{2}. \end{array}$$
(78)$$q_{42}\left(\xi \right)=\frac{A_{1}\sqrt{4d_{0}d_{2}-d_{1}^{2}}\tan\left(\frac{\sqrt{4d_{0}d_{2}-d_{1}^{2}}}{2}\xi \right)-A_{1}\sqrt{d_{1}^{2}-4d_{0}d_{2}}}{2d_{2}}e^{i\left(\alpha_{1}x+\alpha_{2}y+\alpha_{3}z+t\tau +\epsilon \right)},\;4d_{0}d_{2}>d_{1}^{2}.$$

## 5. Modulation Instability

Several nonlinear systems reveal instability that results in the modulation of the steady state owing to the connection among nonlinear and dispersive effects. We examine the MI of model (Equation 1) employing the standard linear stability analysis [3,4,5,41,43]. The solutions of Equation (Equation 1) in the steady-state form are as
(79)$$q\left(x,y,z,t\right)=\left(\sqrt{P_{o}}+A\left(x,y,z,t\right)\right)e^{i\phi (z)},\;\;\phi \left(z\right)=\delta \epsilon P_{o}z,$$
where the $P_{o}$ optical power is normalized. Through using the analysis of linear stability, the perturbation A(x,y,z,t) is examined. Using Equation (Equation 79) in Equation (Equation 1) and linearizing, we get
(80)$$\begin{array}{c} 2i\frac{\partial A}{\partial z}+\left(1-2i\eta \right)\left(\frac{\partial^{2}A}{\partial x^{2}}+\frac{\partial^{2}A}{\partial y^{2}}+\frac{\partial^{2}A}{\partial t^{2}}\right)+2\left(i\gamma +2P_{o}-\delta \epsilon P_{o}-2i\lambda P_{o}+3i\mu P_{o}^{2}-3\nu P_{o}^{2}\right)A\\ +2P_{o}\left(1-i\lambda +2i\mu P_{o}-2\nu P_{o}\right)A^{*}=0. \end{array}$$
The above Equation (Equation 80) can be resolved easily in wave number domain. However, as $A^{*}$ terms (which shows a complex conjugate), the Fourier terms at ω and −ω are coupled so we seek the solution of Equation (Equation 80) that has a form as follows:(81)$$\begin{array}{c} A\left(x,y,z,t\right)=\alpha_{1}e^{i(k_{1}x+k_{2}y+k_{3}z-\omega t)}+\alpha_{2}e^{-i(k_{1}x+k_{2}y+k_{3}z-\omega t)}, \end{array}$$
where $k_{1}$ is normalized wave number and and ω is a frequency of A(x,y,z,t). Substituting Equation (Equation 81) in Equation (Equation 80), we obtained a dispersion relation as follows:(82)$$k_{3}=\pm \frac{\sqrt{-B\left(\left(2\eta +i\right)\left(k_{1}^{2}+k_{2}^{2}+\omega^{2}\right)+8i\gamma \delta \lambda \epsilon P_{o}^{3}+2\left(\lambda +i\right)P_{o}-18\mu \nu P_{o}^{4}-4\left(\mu +i\nu \right)P_{o}^{2}\right)}}{2}.$$
where $B=\left(2\eta +i\right)\left(k_{1}^{2}+k_{2}^{2}+\omega^{2}\right)+8i\gamma \delta \lambda \epsilon P_{o}^{3}-2\left(\lambda +i\right)P_{o}-18\mu \nu P_{o}^{4}+4\left(\mu +i\nu \right)P_{o}^{2}.$ The above dispersion relation (Equation 82) illustrates that stability of steady state be contingent on self-phase and Raman scattering. $-B\left(\left(2\eta +i\right)\left(k_{1}^{2}+k_{2}^{2}+\omega^{2}\right)+8i\gamma \delta \lambda \epsilon P_{o}^{3}+2\left(\lambda +i\right)P_{o}-18\mu \nu P_{o}^{4}-4\left(\mu +i\nu \right)P_{o}^{2}\right)>0$, which shows that the steady state is stable against slight perturbations. In other cases, it becomes unstable. If $-B\left(\left(2\eta +i\right)\left(k_{1}^{2}+k_{2}^{2}+\omega^{2}\right)+8i\gamma \delta \lambda \epsilon P_{o}^{3}+2\left(\lambda +i\right)P_{o}-18\mu \nu P_{o}^{4}-4\left(\mu +i\nu \right)P_{o}^{2}\right)<0$, it shows that $k_{3}$ is imaginary; meanwhile, the perturbation cultivates exponentially. One can see this straightforwardly for instances of MI when $-B\left(\left(2\eta +i\right)\left(k_{1}^{2}+k_{2}^{2}+\omega^{2}\right)+8i\gamma \delta \lambda \epsilon P_{o}^{3}+2\left(\lambda +i\right)P_{o}-18\mu \nu P_{o}^{4}-4\left(\mu +i\nu \right)P_{o}^{2}\right)<0$. Underneath this condition, the evolution rate of the MI gain spectrum h(k) can be expressed as
(83)$$\begin{array}{cc} h(k_{1},k_{2}, & \omega )=2Img\left(k_{1},k_{2},\omega \right)\\ \; & =\sqrt{-B\left(\left(2\eta +i\right)\left(k_{1}^{2}+k_{2}^{2}+\omega^{2}\right)+8i\gamma \delta \lambda \epsilon P_{o}^{3}+2\left(\lambda +i\right)P_{o}-18\mu \nu P_{o}^{4}-4\left(\mu +i\nu \right)P_{o}^{2}\right)}. \end{array}$$

## 6. Discussion of Results and Their Physical Interpretation

Many achieved results of Equation (Equation 1) via a proposed modified extended simple equation, exp(−ϕ(ξ))-expansion, and proposed F-expansion methods that are novel and dissimilar from the constructed results of other techniques [21,30,42]. The main key point to obtain novel results of our proposed methods is the main body of new proposed solutions (Equation 5), (Equation 9), (Equation 16) and ODEs (Equation 6), (Equation 10), (Equation 17). Equations (Equation 6), (Equation 10) and (Equation 17) provide a distinct type of results—for example, trigonometric, hyperbolic trigonometric, rational functions and other solutions through giving dissimilar values of parameters in these ODEs. The exact wave results in the forms of dark-bright solitons, breather-type and multi solitons, kink and anti-kink waves, and periodic and trigonometric function solutions are achieved. The constructed solutions are exact and more general. The authors in [21] discussed the global existence and small dispersion limit and the authors in [30] constructed the approximate solution of this model. The researchers in [42] simulated the vortex tori solitons of this dynamical model. Consequently, our achieved results are innovative and have not been articulated previously.

We demonstrated three-dimensional and two-dimensional structures of some obtained results for the model (Equation 1). In order to observe the physical appearance of this model, the physical structures are described by giving appropriate values to the parameters. In Figure 1, the structures of results (Equation 26)–(Equation 28) are depicted at dissimilar values of parameters: Figure 1A is a multi bright-dark soliton and its 2-dim is in Figure 1B, Figure 1C is a multi solitons interaction and its 2-dim is in Figure 1D, Figure 1E is a periodic soliton and its 2-dim is in Figure 1F. In Figure 2, the structures of results (Equation 32)–(Equation 34) are depicted at dissimilar values of parameters: Figure 2A is a dark soliton and its 2-dim is in Figure 2B, Figure 2C is a anti-kink soliton and its 2-dim is in Figure 2D, and Figure 1E is a Bright soliton and and its 2-dim is in Figure 1F. In Figure 3, the structures of results (Equation 49) and (Equation 50) are depicted at dissimilar values of parameters: Figure 3A is a periodic solitary wave and its 2-dim is in Figure 3B, Figure 3C is a breather-type waves of strange structure and its 2-dim is in Figure 3D.

In Figure 4, the structures of results (Equation 57) and (Equation 59) are depicted at dissimilar values of parameters: Figure 4A is a bright solitary wave and its 2-dim is in Figure 4B, Figure 4C is an anti-kink soliton and its 2-dim is in Figure 4D. The structures in Figure 5 are depicted of results (Equation 64) and (Equation 67) at dissimilar values of parameters: Figure 5A is a dark compact type soliton and its 2-dim is in Figure 5B, Figure 5C is a compact type soliton and its 2-dim is in Figure 5D. In Figure 6, the structures are depicted of results (Equation 73) and (Equation 78) at dissimilar values of parameters: Figure 6A is a periodic soliton and its 2-dim is in Figure 6B, Figure 6C is a multi-kink type solitary wave and its 2-dim is in Figure 6D. Figure 7 is the shape of dispersion relation.

## 7. Conclusions

Three analytical methods have been successfully employed for (3 + 1)-dimension cubic-quintic complex Ginzburg–Landau equation and bright-dark solitons, breather-type and multi solitons interaction, kink and anti-kink solitons, and periodic and other solutions are constructed. The soliton profile is sensitive to entropy, i.e., due to the changes in the entropy amplitude and the width of solitons. The propagation of ultrashort optical solitons in optical fiber is modeled by this equation. The complex Ginzburg–Landau equation with broken phase symmetry has strict positive space–time entropy for an open set of parameter values. The motivation and purpose of this paper is to provide analytical methods to explore exact solutions which helps physicians, mathematicians, and engineers to understand the physical phenomena of this model. The obtained solutions of this article are very helpful in governing solitons dynamics. The traveling wave solutions can also be attained, which are constructed from existing techniques by giving special values to parameters involved in the methods. The model stability is examined by employing the modulation instability analysis, which shows that the model is stable. The computation work and constructed exact solutions endorse the easiness, effectiveness, and influence of the current techniques. These powerful techniques can be employed for several other nonlinear complex PDEs that are arising in mathematical physics.

## Figures and Tables

**Figure 1 entropy-22-00202-f001:**
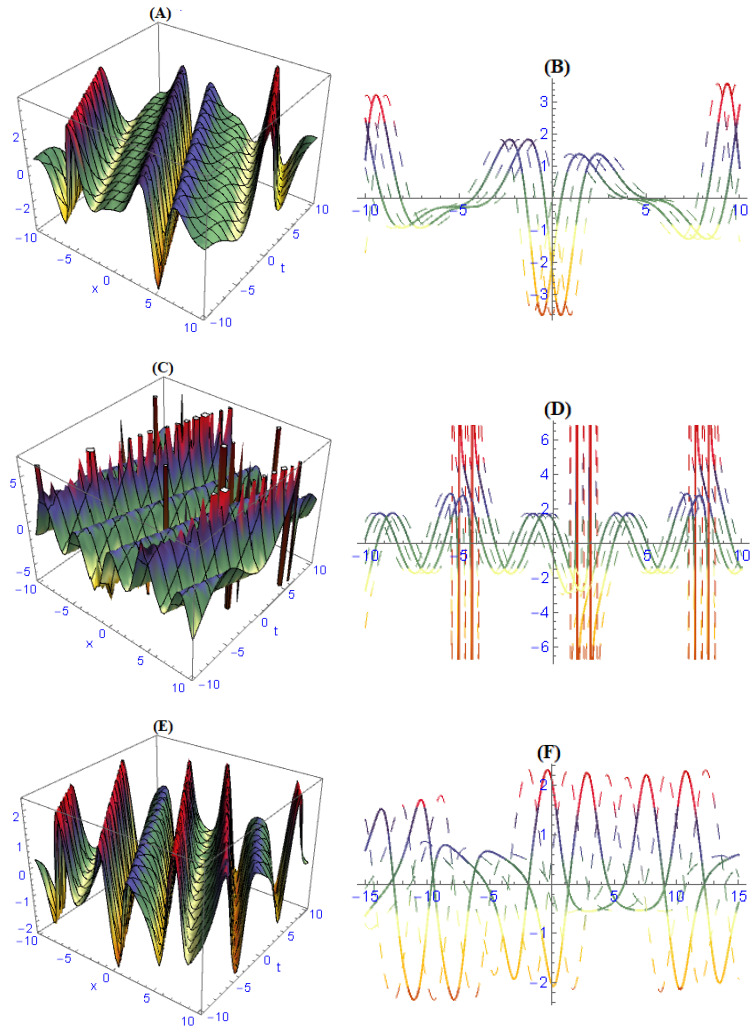
The graph of results (Equation 26)–(Equation 28) are depicted at dissimilar values of parameters and we obtained: (**A**) multi bright-dark solitons and its 2-dim in (**B**), (**C**) multi solitons interaction and its 2-dim in (**D**), (**E**) periodic solitons and and its 2-dim in (**F**).

**Figure 2 entropy-22-00202-f002:**
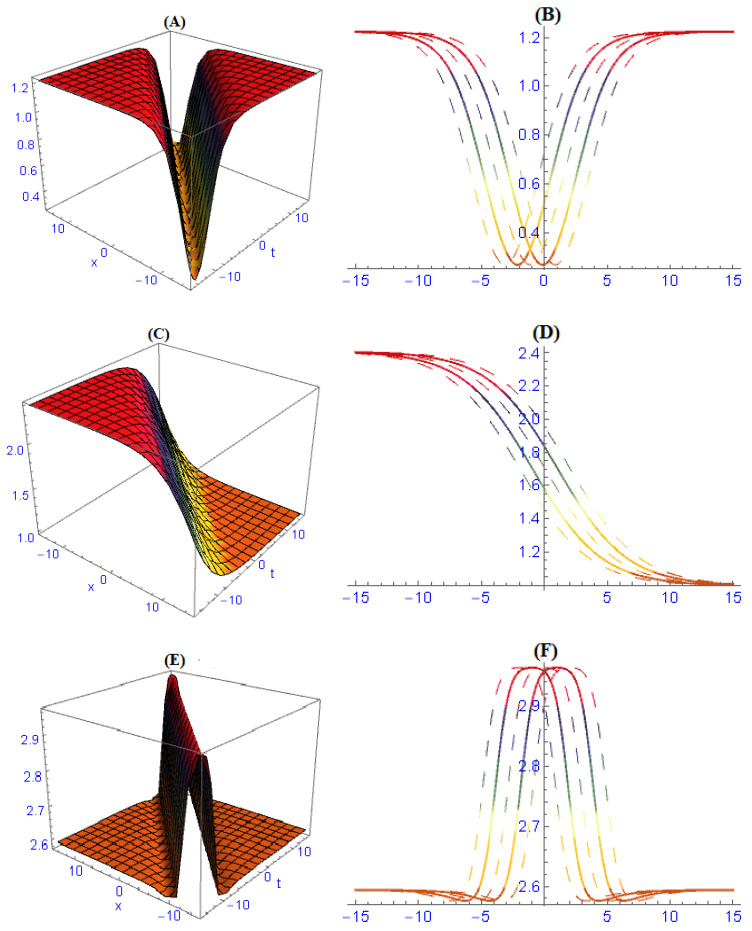
The graph of results (Equation 32)–(Equation 34) are depicted at dissimilar values of parameters and we obtained: (**A**) dark soliton and its 2-dim in (**B**), (**C**) anti-kink soliton and its 2-dim in (**D**), (**E**) bright soliton and and its 2-dim in (**F**).

**Figure 3 entropy-22-00202-f003:**
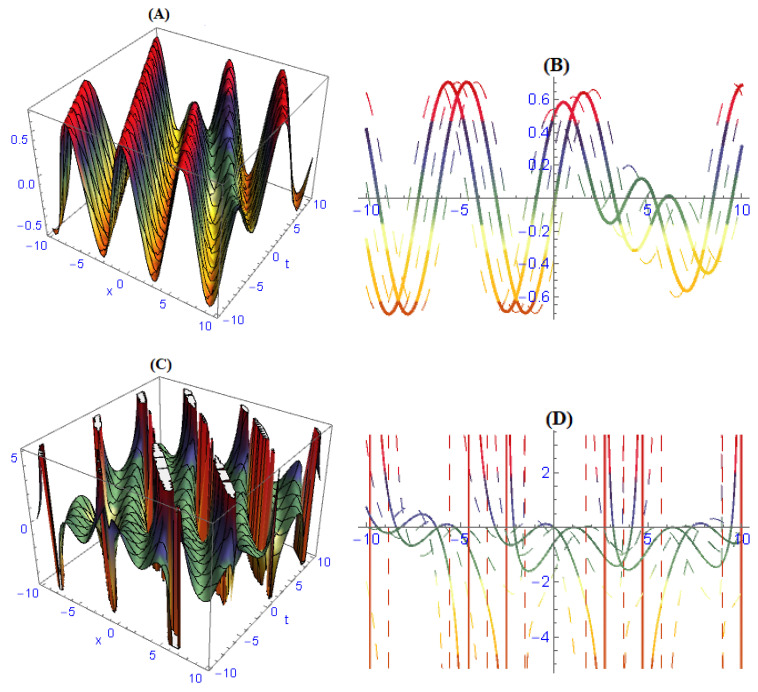
Solitary wave and soliton in various silhouettes are depicted of solutions (Equation 49) and (Equation 50) by choosing different values of parameters: (**A**) periodic solitary wave and its 2-dim in (**B**), (**C**) breather-type waves of strange structure and its 2-dim in (**D**).

**Figure 4 entropy-22-00202-f004:**
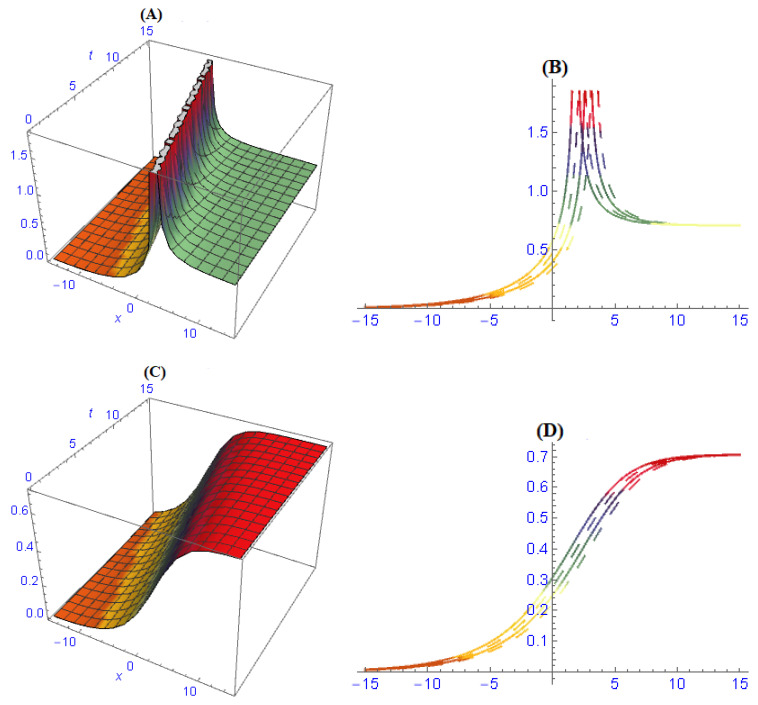
The graph of results (Equation 57) and (Equation 59) are depicted at dissimilar values of parameters and we obtained: (**A**) bright solitary wave and its 2-dim in (**B**), (**C**) anti-kink soliton and its 2-dim in (**D**).

**Figure 5 entropy-22-00202-f005:**
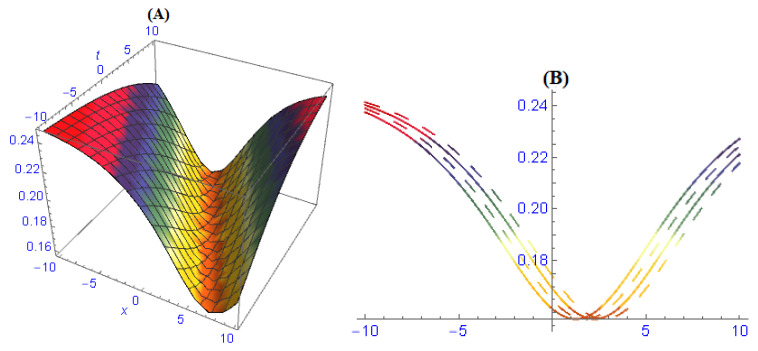
The structures of results (Equation 64) and (Equation 67) are depicted at dissimilar values of parameters: (**A**) dark compact type soliton and its 2-dim is in (**B**), (**C**), compact type soliton and its 2-dim is in (**D**).

**Figure 6 entropy-22-00202-f006:**
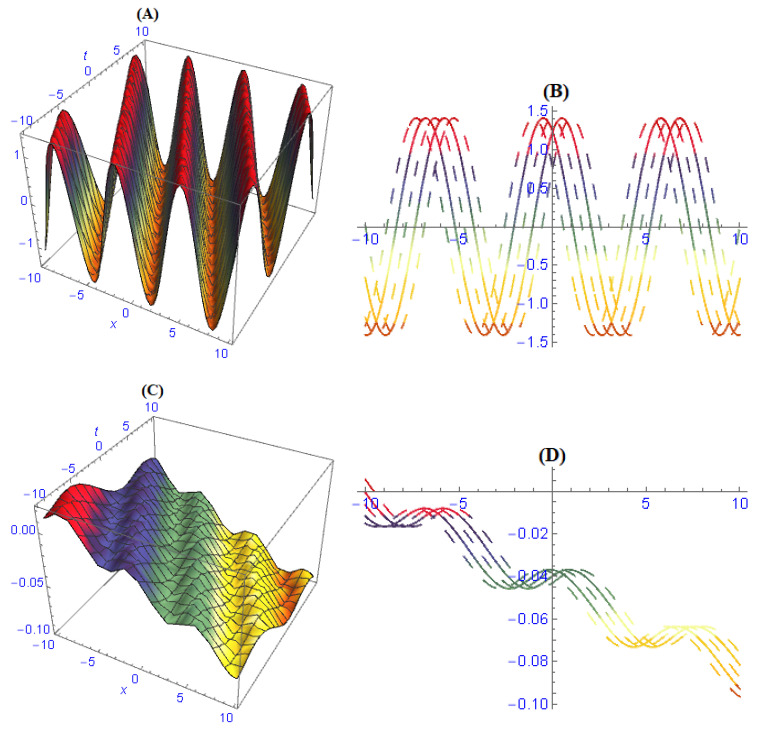
The structures of results (Equation 73) and (Equation 78) are depicted at dissimilar values of parameters: (**A**) periodic soliton and its 2-dim is in (**B**), (**C**) multi-kink type solitary wave and its 2-dim is in (**D**).

**Figure 7 entropy-22-00202-f007:**
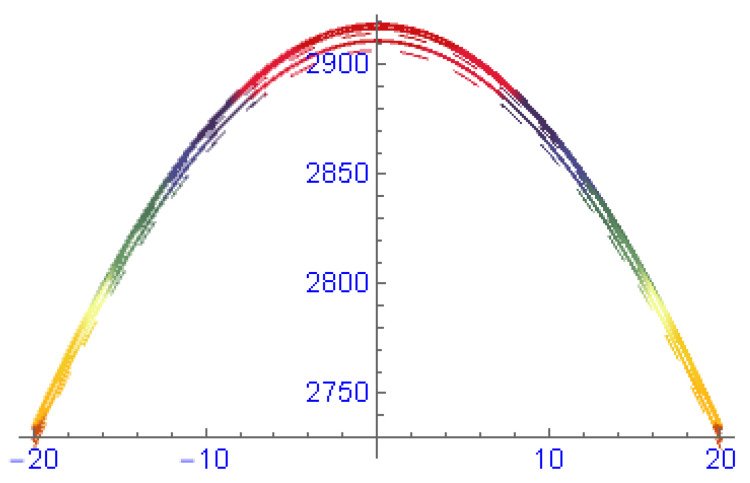
The graph of dispersion relation $k_{3}=k_{3}\left(k_{1},k_{2},\omega \right)$.

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
