# Peer review of "Bright-Dark and Multi Solitons Solutions of (3 + 1)-Dimensional Cubic-Quintic Complex Ginzburg–Landau Dynamical Equation with Applications and Stability"

_entropy, 2020, doi:10.3390/e22020202_

Round 1

Reviewer 1 Report

The Authors have not addressed the comment I raised in my previous review about the fact that the manuscript doesn't match the aims and scope of Entropy. The word entropy is not mentioned in the revised version of the manuscript. This reviewer believes that this manuscript is not suited for publication at Entropy, since it doesn't match the scope of the journal.

Author Response

Response:

The abstract, introduction and conclusion are improved according to suggestion of honorable referee comments and some new references are added.

I read the whole paper carefully and remove the some typing and grammar errors.

Thanks the reviewers for their valuable comments to improve our manuscript.

Reviewer 2 Report

Authors have performed a relevant revision and the manuscript can be recommended for publication in the present form. 

Author Response

Response:

 We carefully checked our revised manuscript, typing errors pointed by the honorable referee and all other errors are corrected.

Thanks the reviewers for their valuable comments to improve and recommended our manuscript for publications.

Round 2

Reviewer 1 Report

The rervised version of the manuscript can be accepted for publication at Entropy, since the Authors have provided links with the aims and scope of the Journal.

This manuscript is a resubmission of an earlier submission. The following is a list of the peer review reports and author responses from that submission.

Round 1

Reviewer 1 Report

This manuscript deals with the construction of solutions for the CQCGL wave equation through different approaches. It presents a variety of soliton-type solutions to the examined equation and discusses the stability properties of such solutions.

This reviewer believes that the manuscript doesn’t match the aims and cope of Entropy, since it fails to satisfactorily demonstrate the “entropic” character of the presented matter. It seems better suited for a different journal, which deals with numerical methods for dynamic problems.

In addition the following points need to be fully addressed if the authors wish to deal with a resubmission of the manuscript, and the English needs to be accurately reviewed, as well.

Abstract:

The initial sentence is too long and involved (line 1-5) We can attain ……………. (line 6) or have attained???? Graphical depiction … (line 10) of what and where????

Introduction:

In the current piece …. (line 59) Piece???? This paper is prepared …. (line 65) Prepared??? (lines 65-69) need to be restructured

Section 6. Discussion (line 116) It needs to be better linked to the remaining sections

Section 7. this section is rather involved and needs to be rewritten in more clear form (lines 118-136)

Section 8. Conclusion

The obtained solutions of this article ………..(line 116) or mentioned in this article?? Which computation work????(line 146) current techniques???? (line 147)

Reviewer 2 Report

In general, this study falls into a wide research direction focused on the existence of solitary waves in nonlinear systems described by PDEs. However, authors do follow the straightforward approach which has had attracted a lot of criticism. Such an approach can be described by the following steps:

Transform the original PDE to an ODE by performing the travelling coordinate substitution.  Balance the highest orders of the nonlinear terms and the derivatives.  Propose (guess) the structure of the solution.  Use computer algebra to identify the parameters of the solution.  Check if the derived solution does satisfy the ODE.

The main criticism for such an approach is focused on the step number 3. It has been demonstrated in many different comments that proposing the structure of the solution (and subsequent identification of parameters) may yield wrong results. A typical reference highlighting this point:

Comments on "A new algorithm for automatic computation of solitary wave solutions to nonlinear partial differential equations based on the Exp-function method". Applied Mathematics and Computations. (2014) vol.243, p.419-425.

Authors should comment these issues in the revised paper because they do propose the structure of the solution (without performing the direct and the inverse balancing of parameters). Moreover, authors should not only add appropriate comments, but also should check if the derived solutions do satisfy their ODEs. 

The second part of the criticism is focused on the fact, that the described approach may yield solitary solutions, - however such an approach cannot produce necessary and sufficient conditions of the existence of these solutions in the space of system's parameters and initial conditions. 

It has been shown in the literature, that proper application of the inverse balancing technique with the operator method can automatically yield the structure of the solution (without any need to guess this structure) - and also does produce the necessary and sufficient conditions of the existence of the constructed solution. A good example on these issues:

Algebraic operator method for the construction of solitary solutions to nonlinear differential equations. Communications in Nonlinear Science and Numerical Simulation. (2013) vol.18(6), p.1374-1389.

Authors should discuss these issues in the revised paper.

Authors also aim to investigate higher order solitary solutions (multi-hump solitons). Authors should be aware that such a straightforward approach as used in their paper can also yield wrong results. An appropriate framework for the construction of multi-hump solitary solutions is given here:

An analytical scheme for the analysis of multi-hump solitons. Advances in Complex Sciences. (2019) vol.22(01), article no.1850027.

Authors should discuss these issues in the revised paper and should check if their higher-order solitary solutions do satisfy the ODEs. 

Finally, authors should provide a clear motivation why this study should be in the scope of this Journal. 

A major revision is recommended.